# Infusion of Some but Not All Types of Human Perinatal Stromal Cells Prevent Organ Fibrosis in a Humanized Graft versus Host Disease Murine Model

**DOI:** 10.3390/biomedicines11020415

**Published:** 2023-01-31

**Authors:** Ramon E. Coronado, Elena Stavenschi Toth, Maria Somaraki-Cormier, Naveen Krishnegowda, Shatha Dallo

**Affiliations:** 1Signature Biologics, 4040 W Royal Lane, Suite 100, Irving, TX 75063, USA; 2Department of Obstetrics and Gynecology, Baylor College of Medicine, Houston, TX 77030, USA; 3Transplant Center, UT Health San Antonio, 7703 Floyd Curl Drive, San Antonio, TX 78229, USA; 4Crown Scientific, 3463 Magic Dr, Suite 315, San Antonio, TX 78229, USA

**Keywords:** mesenchymal stem cells (MSCs), graft versus host disease (GvHD), inflammatory diseases, cell therapy, amnion-derived stromal cells, placenta-derived stromal cells, Wharton’s jelly-derived stromal cells, immunoregulation, fibrosis

## Abstract

Allogeneic transplant rejection represents a medical complication that leads to high morbidity and mortality rates. There are no treatments to effectively prevent fibrosis; however, there is great interest in evaluating the use of perinatal mesenchymal stromal cells (MSCs) and other MSCs to prevent fibrosis associated with chronic rejection. In this study, we isolated human perinatal stromal cells (PSCs) from amnion (AM-PSC), placental villi (PV-PSC), and umbilical cord (UC-PSC) tissues, demonstrating the phenotypic characteristics of MSCs as well as a >70% expression of the immunomodulatory markers CD273 and CD210. The administration of a single dose (250,000 cells) of each type of PSC in a humanized graft versus host disease (hGvHD) NSG^®^ murine model delayed the progression of the disease as displayed by weight loss and GvHD scores ranging at various levels without affecting the hCD3+ population. However, only PV-PSCs demonstrated an increased survival rate of 50% at the end of the study. Furthermore, a histopathological evaluation showed that only PV-PSC cells could reduce human CD45+ cell infiltration and the fibrosis of the lungs and liver. These findings indicate that not all PSCs have similar therapeutic potential, and that PV-PSC as a cell therapeutic may have an advantage for targeting fibrosis related to allograft rejection.

## 1. Introduction

Allogeneic transplant rejection represents a medical complication that leads to high morbidity and mortality rates. The condition is classified as acute (short term) or chronic (long term). Acute transplant rejection is less common and can be managed due to advances in broad spectrum immunosuppressants. In contrast, chronic rejection has no adequate treatment, and it represents an important unmet clinical need. In the case of hematopoietic stem cell transplantation, rejection is characterized when donor cells recognize the recipient’s cells as “non-self” and engage in a broad attack against host tissues in a process known as graft versus host disease (GvHD). Chronic graft versus host disease (cGvHD) is estimated to affect up to 50% of patients within 5 years of receiving the allogeneic transplant [1]. Clinical cGvHD includes characteristics of acute rejection in addition to more diverse components that resemble autoimmune syndromes, such as progressive allograft injury, which is primarily characterized by obliterative arteriopathy and interstitial fibrosis [2,3]. For example, cGvHD of the lungs can manifest as bronchiolitis obliterans accompanied with airflow obstruction [4]. Similarly, cGvHD in the liver has been associated with hepatic ductopenia and fibrosis [5]. Because cGvHD allograft rejection shares many of the pathologies associated with autoimmune disorders, therapies that promote immunoregulation and immune tolerance are of great interest.

While the reported immunoregulatory activity of mesenchymal stromal cells (MSCs), notably their ability to control aberrant inflammatory responses, has positioned them as a promising option for the treatment of cGvHD, the clinical translation of this type of therapy in over 70 trials across the globe have demonstrated varying degrees of efficacy [6]. The donor tissue source from which MSC therapy is manufactured is a contested factor with no consensus within the GvHD field. Bone marrow mesenchymal stem cells (BM-MSCs) and adipose-derived mesenchymal stem cells (AD-MSCs) have been studied in 61% of GvHD clinical trials to date, with the age of the donor cells being identified as a predictive factor for patient survival [7]. Stem cells isolated from birth tissues, collectively known as perinatal stromal cells (PSCs), are similar to MSCs in terms of therapeutic potential [8] but offer several advantages related to the age of cell source (at birth), ease of acquisition, and non-invasive collection, as afterbirth tissue is considered medical waste. Furthermore, PSCs are gaining interest in the cell therapy field, with the intention of leveraging their purported innate immune regulation functions between the mother and the fetus during normal pregnancy (i.e., immune tolerance, anti-inflammation, etc.) [9,10,11,12]. PSCs also elicit immunoregulatory properties and capabilities to modulate organ fibrosis, a complication of GvHD [13,14,15]. Clinical evidence has shown promise with respect to using PSCs for acute GvHD (aGvHD); the administration of placenta-derived decidual stromal cells therapy improved the 1-year survival rates as compared with bone marrow MSCs [16,17]. Similarly, the repeated infusion of umbilical cord PSCs has been shown to reduce the incidence of cGvHD development in haplo-identical hematopoietic stem cell transplantation [4]. However, the therapeutic effect of PSCs for the treatment of GvHD may differ depending on the various post partum tissue from which they are derived, such as the umbilical cord (UC), Wharton’s jelly (WJ), amnion, chorion, and placenta.

Here, we evaluate and compare the therapeutic capability of PSCs derived from three different perinatal sources, amnion perinatal stromal cells (AM-PSC), placenta villi perinatal stromal cells (PV-PSC), and Wharton’s jelly perinatal stromal cells (WJ-PSC), in their ability to ameliorate disease progression in a humanized GvHD NSG^®^ murine model. These tissue sources were selected based on the most frequently studied tissue sources in research and clinical studies. The progression and severity of GvHD was evaluated by recording the weight loss, which is representative of the health of the mouse, GvHD score [18], survival, proportion of human CD45+ lymphocyte infiltration, and histopathology fibrotic scoring in the lung (Ashcroft) and liver (Ishak).

## 2. Materials and Methods

### 2.1. Cell Preparation

Healthy, full-term placentas were obtained from the Cooperative Human Tissue Network, which is funded by the National Cancer Institute; these were obtained after maternal signed informed consent, which was approved by the University of Alabama Birmingham Institutional Review Board under protocol #940831016 in accordance with relevant guidelines and regulations. Human placental tissues were processed within 24 h of collection in a sterile laminar hood. First, the amnion membrane was mechanically separated from the chorion and extensively washed with phosphate-buffered saline (PBS). Afterwards, it was minced into small pieces and digested with TryPLE (Gibco, Waltham, MA, USA) at a rate of 5 mL/g of tissue for 30 min in a shaker incubator (I24 Incubator Shaker series, New Brunswick Scientific, Edison, NJ, USA) at 37 °C and 150 rpm to remove the amniotic epithelial cells. The undigested amnion was then removed, washed with PBS, and further digested with 125 U/mg of collagenase I (Worthington, Lakewood, NJ, USA) at 37 °C and 150 rpm for 1.5 h to isolate the AM-PSC. The mobilized cells in the digest were passed through a 100 µm cell strainer (VWR, Radnor, PA, USA) and collected by centrifugation at 500× *g* for 8 min. Second, the WJ-PSC was extracted from the umbilical cord as follows: first, the umbilical cord was sectioned into approximately 1.5 cm long pieces, which were then longitudinally dissected to expose the Wharton’s jelly. The arteries and vein were removed, and the remaining tissue was minced into small pieces and digested with 125 U/mg of collagenase I at 37 °C and 150 rpm for 2.5 h or until all tissue was digested. The digest was passed through a 100 µm cell strainer and centrifuged at 500× *g* for 8 min. For the isolation of PV-PSCs, the placenta was thoroughly washed with PBS, minced into small pieces, and digested with 125 U/mg of collagenase I at 37 °C and 150 rpm for 1.5 h. Afterwards, the tissue digest was passed through a 100 µm cell strainer and centrifuged at 500× *g* for 10 min to collect the PV-PSCs.

### 2.2. Culture of Isolated Perinatal Stromal Cells

Freshly isolated PSCs were cultured under standard tissue culture conditions (humidified, 37 °C, and 5% CO_2_) in MEM-alpha (Gibco, Waltham, MA, USA) supplemented with 1% antibiotic–antimycotic (Gibco) and 5% heat-inactivated FBS (Gibco). The cell culture medium was replaced every other day, and cells were sub-cultured when they reached 70–80% confluency.

### 2.3. Mice

Non-obese diabetic mice with severe combined immunodeficient-IL-2 receptor gamma-null (NOD.Cg-Prkdc^scid^ Il2rg^tm1Wjl^/SzJ, NSG^®^) were purchased from Jackson Laboratories (Bar Harbor, ME, USA). The mice were contained in a Charles River animal care facility, with food and water made available under pathogen-free conditions. The mice were exposed to a 12:12 h light/dark cycle at a controlled room temperature and humidity. All animal experiments protocols were conducted with the approval of Charles River’s Institutional Animal Care and Use Committee in accordance with the relevant guidelines and regulations.

### 2.4. GvHD Induction and Treatment

Xenogeneic GvHD was generated in 6–8-week-old female NSG^®^ mice, which were engrafted intravenously (IV) via tail vein injection with 3 × 10^7^ human peripheral blood mononucleated cells (PBMCs) and sorted into five groups of ten animals each. A positive control (PBMC) group received no treatment to allow for GvHD development. A negative control (CsA) group was administered with cyclosporin A, a known immunosuppressant used to prevent transplant rejection; the administration was conducted once daily intraperitoneally (IP) at 15 mg/kg from Day 1 to study end to repress GvHD symptoms. On Day 5, three groups of treated mice received PSC infusions at a single dose of 250,000 of cells in 100 µL of PBS, which was delivered via intravenous tail vein injection (Figure 1A). The injection was prepared as follows: cells were recovered from cryopreservation and allowed to recover in an incubator (humidified, 37 °C, and 5% CO_2_) for 48 h before injection. Afterwards, the PSCs were detached using 0.25% trypsin (Gibco), neutralized with trypsin neutralizer solution (Gibco), collected by centrifugation at 450× *g* for 5 min, and then prepared as a 250,000-cell suspension in 100 µL of PBS. All cells used were between passages 2 and 3. The treatment groups were designated as follows: amnion perinatal stromal cells (AM-PSC), Wharton’s jelly perinatal stromal cells (WJ-PSC), and placental villi stromal cells (PV-SC).

Animal survival was monitored daily with weight and GvHD scoring being assessed three times a week. The clinical GvHD index was scored based on 5 parameters: weight loss, activity, posture, fur texture, and skin integrity. These parameters were placed on a scale ranging from 0 to 2, with 2 being the most severe (Table 1). Animals were euthanized when their weight reached a weight loss of >30% or after two consecutive loss measurements of >15%. Five animals in each group were analyzed at Day 40 for the expression of human CD3+ in whole blood (0.1 mL was collected by mandibular bleeds). Experiments were carried out until Day 55.

The development of GvHD was assessed using established histopathology in Masson’s trichrome-stained slides, which quantitatively scored lung (Ashcroft) and liver (Ishak) fibrosis [19,20]. Briefly, liver and lung (inflated) samples were collected from the animals of each group as they reached endpoint. All organs were preserved in formalin for 24 h, transferred to 70% ethanol, and shipped at room temperature to Histowiz (Brooklyn, NY, USA) for processing to formalin-fixed paraffin-embedded (FFPE) blocks, H&E-stained slides, and special stains (Masson’s Trichrome and human CD45+ staining). Certified pathologists (blinded) contracted by Histowiz provided histopathology scores for liver (Ishak) and lung (Ashcroft) fibrosis, as well as digitally quantified human CD45+ stained cells.

### 2.5. Flow Cytometry

#### 2.5.1. PSC Samples

Cultured PSCs were washed with running buffer (Miltenyl Biotec Inc., Auburn, CA, USA) and centrifuged at 350× *g* for 5 min (Eppendorf, Westbury, NY, USA). The cells were incubated in blocking solution (Blockaid, Thermo, Austin, TX, USA) at 4 °C for 15 min. PSC samples (1 × 10^5^ cells/100 µL) were incubated with the following antibodies: CD85d-ILT4-PE, HLA-DR- TU36-PE, CD45-HI30-Brilliant Violet, CD73-AD2-PE (StemCell Technologies, Vancouver, BC, Canada), CD90- 5E10-PE (Molecular Probes, Eugene, OR, USA), CD105-43A3-PE, CD273-B7DC-PE, CD119-IFNgRa-PE, CD40-5C3-FITC, CD11b-M1/70-FITC, and CD178-NOK-1-PE (Biolegend, San Diego, CA, USA), and HLAG9-MEM-G/11-FITC (Invitrogen, Austin, TX, USA). Upon completion of the incubation, cells were washed twice with running buffer, centrifuged at 350× *g*, and then resuspended in 100 µL of running buffer containing 300 mM of DAPI (Biolegend), where they were incubated at 4 °C for 15 min in the dark. Analysis was performed using a Cytoflex flow cytometer (Beckman Coulter, Irving, TX, USA).

#### 2.5.2. Mice Samples

Whole blood samples were processed by centrifugation, and red blood cells (RBCs) were lysed with ACK buffer according to the manufacturer’s instructions. The final single-cell suspensions were prepared in staining buffer (PBS pH 7.4, 2.5% FBS, 0.09% NaN_3_), and 2 × 10^7^ cells/mL were added into 96-well plates and stained for 30 min at 4 °C with 100 μL of the reconstituted Live/Dead Aqua (Life Technologies), following the manufacturer’s instructions. After two washes with 1 mL of staining buffer, the Fc receptors were blocked using 100 μL of TruStain Fc solution (Biolegend) for 5–10 min on ice prior to immunostaining. Cells were stained for 30 min at 4 °C with antibodies; then, they were washed twice with 1 mL of staining buffer and resuspended in 100 μL of staining buffer for analysis. Isotype control antibodies were used as negative staining controls when deemed necessary. All data were collected on a FortessaLSR (BD) system and analyzed with FlowJo software (Tree Star, Inc., Ashland, OR, USA). Cell populations were defined according to the protocol, and the gating strategy was determined by initial gating on singlets (FSC-H vs. FSCA), then on live cells based on Live/Dead Aqua viability staining. The percentage of human CD3+ (CD3 PE HIT3a, Biolegend) cells was determined according to the parent cell gate.

### 2.6. Statistical Analysis and Imaging

A survival curve followed by a Log-rank (Mantel–Cox) test was utilized for the survival analysis. For other statistical analyses, One-way or two-way ANOVA tests followed by Dunnett’s multiple comparisons test were performed using GraphPad Prism version 8.0.1 (GraphPad Software, San Diego, CA, USA). Normality tests and F tests confirmed the Gaussian distribution and equality of variance between different groups. Values are presented as the mean ± SEM, and differences of *p* < 0.05 were considered statistically significant. The graphical abstract was created with the Biorender.com tool.

## 3. Results

Isolated PSCs from three different perinatal donor tissue presented spindle-like morphology when cultured, and displayed variable levels of surface markers typically found in MSCs such as CD73, CD90, and CD105, as well as a negative or low expression of lymphocytic markers CD11b, CD45, and HLA-DR (Figure 1B). Interestingly, only WJ-PSCs displayed >95% of the MSC phenotype markers CD105, CD73, and CD90 according to the minimum criteria for identifying MSCs as per ISCT guidelines [21]. The variable levels of surface MSC markers of PV-PSCs and AM-PSCs may indicate a heterogeneous population of stromal cells. Cultured PSCs also displayed immune-related markers (Figure 1C), including high CD273 (PD-L2) and CD210 (IL-10R) and <10% or a negative expression of CD178, CD119, CD85d, and CD40.

All placental-derived cell infusions were well-tolerated with no observations of morbidity nor mortality. Cyclosporin A (CsA), a known immunosuppressant drug, was utilized to mitigate the effects of human PBMC transplant rejection and act as an active control for the delay of GvHD disease progression. Human PBMC were successfully engrafted in NSG^®^ mice, which was confirmed by the detection of human CD3^+^ (hCD3+) cells in mice blood at Day 40 (Figure 2A), thereby indicating the presence of GvHD disease (PBMC group). A single administration of 250,000 PSCs on Day 5 of the study did not impact the level of hCD3+ cells as compared with PBMC on Day 40. In contrast, daily administration of cyclosporin A (CsA) significantly reduced the level of hCD3+. GvHD was developed in engrafted mice and detected by weight loss, a primary characteristic of the progression of the disease (Figure 2B). Compared with PBMC, only the CsA and PV-PSC groups displayed a significant difference in percentage weight change over time, indicating a slower disease progression.

All PSCs and CsA treatments delayed the rate of progression of GvHD to some degree when measured as a return to baseline (in days), i.e., the number of days for the weight to return to the Day 0 level (Figure 3A). However, only CsA and PV-PSC demonstrated the most uniform (statistically significant, *p* < 0.05) effect in delaying the disease progression. The GvHD score over time (Figure 3B) and area under the curve (AUC) for the GvHD score at Day 29 and 52 (Figure 3C) for the surviving animals show that the presentation of the disease is only attenuated with WJ-PSC, PV-PSC, and CsA. Interestingly, AM-PSC was found to exacerbate the presentation of GvHD in mice, as seen by the progression of GvHD scores over time and the AUC. These observations are further reflected in the survival rate of the mice at Day 55 of the study, where only PV-PSC out of all PSCs displayed an improved survival rate of 50%. CsA as an active control displayed a 90% survival rate. Most of the animals in the experimental and PBMC groups were euthanized due to severe weight loss (>30%).

Human CD45+ cell infiltration in lung (Figure 4A) and liver (Figure 4B) tissues as a sign of allograft injury related to GvHD disease was evaluated by the quantification of human CD45+ (hCD45+) stained cells (percentage). The majority of hCD45+ cells were identified surrounding the periphery of vascular lumen, which is a pattern typically seen with infiltrating cells from the vascular system. Compared with the PBMC group, only the PV-PSC and active control (CsA) groups showed a significant reduction of infiltrating human CD45+ cells into the tissue, which supports the above observations (GvHD score and weight loss) (Figure 4C,D). Histological sections of mice lung (Figure 5A) and liver (Figure 5B) stained with Masson’s trichrome (collagen depositions were stained blue) indicate signs of tissue remodeling and fibrosis across all groups. The scoring of fibrosis for the lungs (Ashcroft) and liver (Ishak), conducted in a blinded manner by a pathologist, showed that only PV-PSCs significantly decreased fibrosis in the lungs and liver, with WJ-PSC demonstrating reduced fibrosis only in the liver and AM-PSCs having no effect on fibrosis. CsA completely attenuated fibrosis only in the liver.

## 4. Discussion

The perinatal stromal cells (PSCs) isolated in this study had characteristics similar to MSCs based on the ISCT criteria [21], such as being plastic adherent under standard culture conditions, expression of CD105+, CD73+, CD90+, and a negative expression or absence of CD11b-, CD45-, and HLA-DR. Interestingly, the levels of positive expression markers for PV-PSCs and AM-PSCs varied and did not reach >95% as per the ISCT, potentially indicating a heterogenous cell population. Furthermore, their capability to differentiate into tri-lineage (chondrocytes, osteocytes, and adipocytes) was not tested as we theorized that the effect of PSCs in GvHD is based not on their differentiation capabilities (stemness) but rather in their immunomodulatory effects; hence, we referred to them as stromal cells. To identify the potential immunomodulatory effect of PSCs, we tested other immune-pertinent markers [22,23,24]. We determined that all PSCs included in this work were positive (>70%) for CD273 (PD-L2) and CD210 (IL-10 Receptor) and <10% or negative for CD178 (FasL), CD119 (IFNg Receptor), CD85d (ILT4), and CD40. In addition, the presence of immune regulatory receptors could help identify priming/licensing strategies in the future for tailoring a therapeutic index.

This humanized GvHD NSG^®^ mouse model showed that PSCs may delay the disease progression and manifestation as well as improve the survival rate, but this is specific to the type of PSCs used. Only PV-PSCs demonstrated a significant effect on the delay of body weight loss, lower GvHD scores over time, and increased survival rate of 50% by Day 55. In contrast, AM-PSCs demonstrated a worsening of the GvHD as displayed by increased GvHD scores with time, exceeding the PBMC control group and no survival by the end of the study. Although the AUC for the GvHD scores at Day 29 demonstrate that all PSCs have had some positive effect on attenuating the presentation and progression of the disease as compared with the PBMC group, the presence of hCD3+ cells but not CsA at Day 40 in all PSCs groups indicates an active disease state. Furthermore, the fact that the survival rate only increased with PV-PSCs treatment indicates that this is potentially the only cell type out of all PSCs that has a greater effect on disease progression without being curative at a single dose administration. A meta-analysis of MSC use as prophylaxis in animal models of aGvHD showed that the mortality rate decreased with the co-transplantation and day-after transplantation of allogeneic MSCs with BM-MSCs and umbilical cord blood MSCs, but this was not the case for umbilical cord MSCs [25]. Thus, the model may be improved by timing the treatment administration at co-transplantation with hPBMCs for the aGvHD effects or between Day 20 and 30 for the cGvHD effects as a single dose vs. multiple dose administration; this would provide a better insight of which PSCs may be effective as prophylaxis, curative, or not recommended for GvHD.

In this GvHD mouse model, we also looked at the immunological rejection of the transplant by quantifying human CD45+ infiltrates as well as organ fibrosis in the liver and lungs. Treatment with PV-PSC showed a significant reduction in the infiltrating hCD45+ cells as well as decreased fibrotic score changes in the lung and liver; however, the percentage of circulating hCD3+ was not different from the other treatment groups. This suggests that there is a possibility that PV-PSC are capable of changing the migratory, activation, or chemotactic activity of human CD45+ cells. Alternatively, it is also possible that PV-PSC could influence endothelial cells and their capability to hinder activated human CD45+ cell migration. While the in vivo mechanism of action of these hypotheses has not yet been demonstrated here, in vitro studies have shown that PSCs from umbilical cord and placenta villi can significantly reduce CD3+ T-cell proliferation [26,27]. It is rather interesting that these effects were only present with PV-PSCs, thereby highlighting that not all PSCs are the same and that their utility for GvHD should be rigorously evaluated.

## 5. Conclusions

In summary, PSCs from the umbilical cord, placenta villi, and amnion were isolated from within the same donor and characterized to express phenotypical MSCs markers in addition to immunomodulatory markers. The administration of all PSCs was well-tolerated, but not all cell types had the same therapeutic effect. Only PV-PSC cellular therapy demonstrated the potential for efficacy toward GvHD based on various outcome measurements including GvHD score, weight loss, organ fibrosis, and human CD45+ infiltration. However, a better understanding of the mechanism of action, dose response and a multiple dosing procedure would be necessary in order to translate this potential therapeutic effect into clinical applications.

## Figures and Tables

**Figure 1 biomedicines-11-00415-f001:**
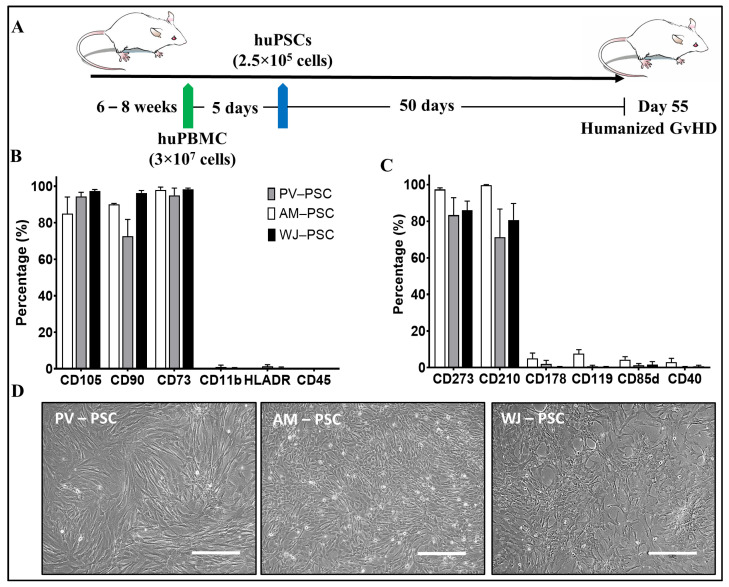
Graft versus host disease (GvHD) model and PSC phenotype. (**A**) Xenogeneic GvHD induction diagram (injection times, etc.). Perinatal Stromal Cell flowcytometry characterization for (**B**) mesenchymal stem cell-related markers and (**C**) immune-related markers. Representative images of PSC in vitro adherent morphology (**D**). *n* = 5 independent donors. Abbreviations: human peripheral blood mononucleated cells (huPBMC), human perinatal stromal cells (huPSCs), amniotic perinatal stromal cells (AM-PSC), placental villi perinatal stromal cells (PV-PSC), and Wharton’s jelly perinatal stromal cells (WJ-PSC). Scale bar = 400 µm.

**Figure 2 biomedicines-11-00415-f002:**
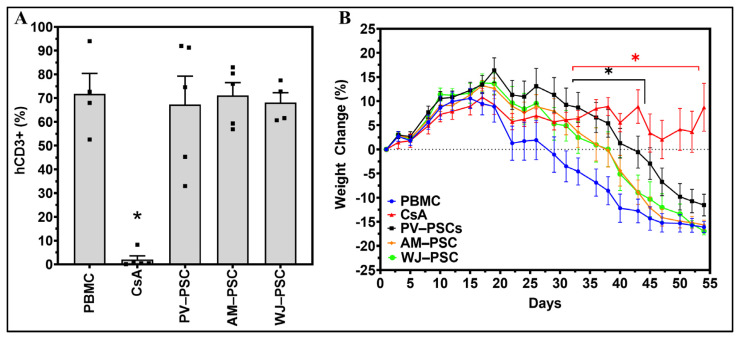
Administration of PV-PSCs delay weight loss in the progression of GvHD. (**A**) Population of human CD3+ present in mice at Day 40 indicative of the presence of the GvHD disease state. (**B**) Daily percent weight change per experimental condition with last carried observation carried over included for weight in cases of animal death. A two-way ANOVA test for the change in weight with time and treatment was performed and denoted as significant at * *p* < 0.05. PBMC and cyclosporin A are the respective positive (GvHD) and negative (attenuated GvHD) controls. Abbreviations: human peripheral blood mononucleated cells (huPBMC), human perinatal stromal cells (huPSCs), amniotic perinatal stromal cells (AM-PSC), placental villi perinatal stromal cells (PV-PSC) and Wharton’s jelly perinatal stromal cells (WJ-PSC).

**Figure 3 biomedicines-11-00415-f003:**
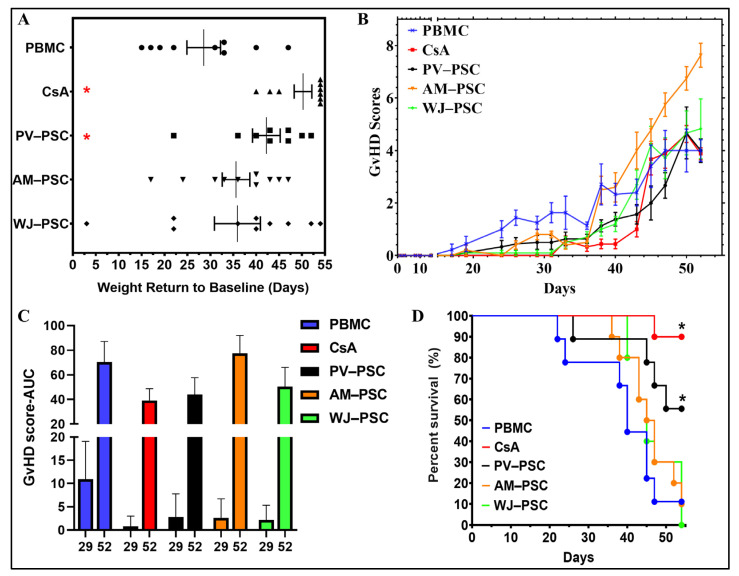
PV-PSC treatment significantly improved the survival rate of GvHD-affected mice as compared with other PSCs. (**A**) Time for weight return to baseline (zero change). (**B**) GvHD progression over the course of the 55-day study, as denoted by the GvHD score of the surviving animals (mean±SEM). (**C**) Area under the curve at Day 29 and 52 of the study for all treatments, indicative of cumulative GvHD progression. (**D**) Survival rate at Day 55. *n* = 9–10 tested in each group. Statistical significance denoted as * *p* < 0.05. Data presented as the average ± SEM. Abbreviations: human peripheral blood mononucleated cells (huPBMC), human perinatal stromal cells (huPSCs), amniotic perinatal stromal cells (AM-PSC), placental villi perinatal stromal cells (PV-PSC) and Wharton’s Jelly perinatal stromal cells (WJ-PSC).

**Figure 4 biomedicines-11-00415-f004:**
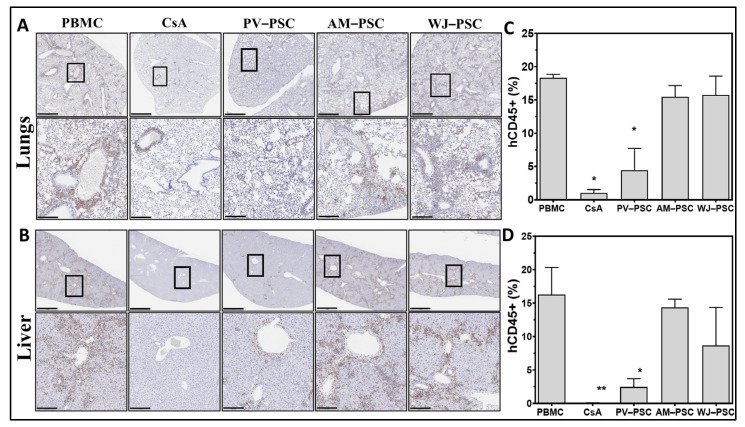
PL-PSC preferentially diminishes human CD45+ cell infiltration in the lungs and liver of GvHD mice as compared with amnion or Wharton’s jelly PSCs. Representative histological images of mice lung (**A**) and liver (**B**) tissues stained for human CD45+ cells. Fibrotic pathological scoring of lung fibrosis performed by the Ashcroft score (**C**) and, liver fibrosis by the Ishak score (**D**). Statistical analysis is represented by a one-way ANOVA with Dunnet’s post hoc test. * *p* < 0.05, ** *p* < 0.001, *n* = 3. Upper row scale = 1 mm; lower row scale = 200 µm. Abbreviations: human peripheral blood mononucleated cells (huPBMC), human perinatal stromal cells (huPSCs), amniotic perinatal stromal cells (AM-PSC), placental villi perinatal stromal cells (PV-PSC) and Wharton’s jelly perinatal stromal cells (WJ-PSC).

**Figure 5 biomedicines-11-00415-f005:**
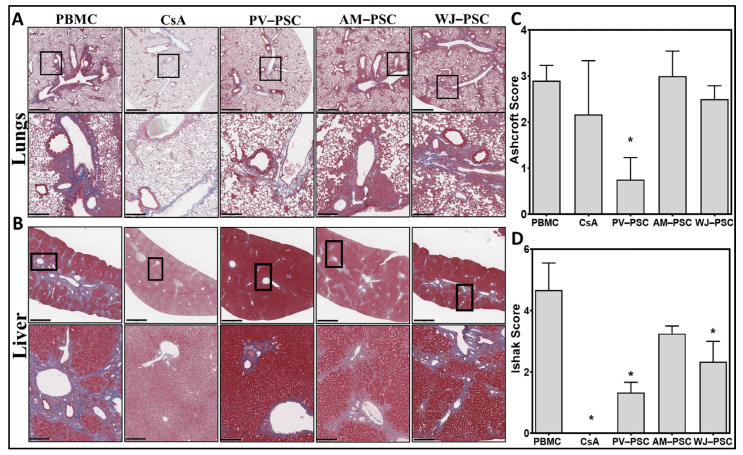
Placenta PSC preferentially diminishes fibrosis in the lungs and liver of GvHD mice as compared with amnion or Wharton’s jelly PSCs. Representative histological images of mice lung (**A**) and liver (**B**) tissues stained with Masson’s trichrome. Fibrotic pathological scoring of lung fibrosis by the Ashcroft score (**C**) and, liver fibrosis by the Ishak score (**D**). Statistical analysis is represented by a one-way ANOVA with Dunnet’s post hoc test. * *p* < 0.05, *n* = 3. Upper row scale = 1 mm; lower row scale = 200 µm. Abbreviations: human peripheral blood mononucleated cells (huPBMC), human perinatal stromal cells (huPSCs), amniotic perinatal stromal cells (AM-PSC), placental villi perinatal stromal cells (PV-PSC) and Wharton’s jelly perinatal stromal cells (WJ-PSC).

**Table 1 biomedicines-11-00415-t001:** Graft versus host disease scoring system.

Criteria	Grade 0	Grade 1	Grade 2
Weight loss	<10%	10%, <25%	25%
Activity	Normal	Mild to moderately decreased	Stationary until stimulated
Posture	Normal	Hunching only at rest	Severe gait, impaired movement
Fur texture	Normal	Mild to moderate ruffling	Severe ruffling/poor grooming
Skin integrity	Normal	Scaling of paws and/or tail	Obvious areas of denuded skin

## Data Availability

The datasets generated during and/or analyzed during the current study are available from the corresponding author upon reasonable request.

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
