# Peer review of "Infusion of Some but Not All Types of Human Perinatal Stromal Cells Prevent Organ Fibrosis in a Humanized Graft versus Host Disease Murine Model"

_biomedicines, 2023, doi:10.3390/biomedicines11020415_

Round 1

Reviewer 1 Report

The manuscript "Infusion of some but not all types of human perinatal stromal cells prevent organ fibrosis in a humanized Graft versus Host Disease murine model", written by Coronado RE, Stavenschi Toth E, Somaraki-Cormier M, Krishnegowda N and Dallo S, presents an experiment in which graft versus host disease in murine model was treated with several types of human perinatal stromal cells. Immunodeficient mice were injected with human peripheral blood mononuclears and followed for up to 55 days. Animals were treated with amniotic, placental villi and Wharton's jelly perinatal stromal cells, as well as with cyclosporine A. Several parameters were measured: animal weight change, number of human CD3+ cells, GvHD score, survival, CD45+ infiltration into lungs and liver etc. Only placental villi cells' treated mice showed improvement in comparison with nontreated controls.

The manuscript is well written and has scientific value. Introduction contains all the necessary data, methods are detailed, and results are clearly presented and justify the conclusion. References are adequate.

Minor comment

On Figure 1, D presents cell morphology, not immune-related markers.

Figures 2, 3, and 4 do not have an explanation of abbreviations

Author Response

On Figure 1, D presents cell morphology, not immune-related markers.

Changed the numeration to reflect the highlighted error. Additionally, added “Representative images of PSC in vitro adherent morphology (d).”

Figures 2, 3, and 4 do not have an explanation of abbreviations

Added explanation of abbreviations to the captions of Figures 2, 3, 4 and 5.

Reviewer 2 Report

In this work, the authors characterized and compared the therapeutic capabilities of AM-PSC, PV-PSC and WJ-PSC from the same donor for GvHD, providing powerful information for clinical treatment of GvHD by PSCs. Below are some questions/comments.

Issues:

Page 5, Results part: Cultured PSCs displayed immune-related markers differentially, such as high expression of CD273 and CD210 but low expression of CD178, CD119, CD85d and CD40. Do authors have any explanation for this phenomenon?

Page 6, first paragraph: Please indicate the function of cyclosporin A in the main text, such as immunosuppressant medication, to help the readers better understand the experimental design.

Figure 2B: Why the weight of mice will increase firstly then decrease after the treatments of all PSCs and CsA? Do the authors have any explanation?

Figure 3A: What do the p-values mean in the figure? AM-PSC and PV-PSC can delay the time for the weight to return to Day 0 level more significantly?

Minor issues:

Page 5, Results part: Please change “Figure 1B & C” to “Figure 1B” and change “Figure 1D” to “Figure 1C”. Please change “>10%” to “< 10%”.

Figure 3: I would recommend using the same order of different experimental groups for the legends in different panels, such as, “PBMC, CsA, PV-PSC, WJ-PSC, AM-PSC”.

Author Response

Page 5, Results part: Cultured PSCs displayed immune-related markers differentially, such as high expression of CD273 and CD210 but low expression of CD178, CD119, CD85d and CD40. Do authors have any explanation for this phenomenon?

This is a first attempt at expanding the characterization of the immune phenotype profile of these cells across all the three tissues types (amnion, Wharton’s Jelly and placenta) within the same donor, for five donors, beyond the standard MSC phenotyping. Currently, we do not have an explanation of this phenomenon, whether it is related to perinatal tissue physiology/anatomy, method of isolation and/or expansion methodology. However, by providing this type of characterization, the intent is to foster research related to the biological and clinical relevance of the immunophenotypic profile of PSCs as a cell therapy for autoimmune conditions.

Page 6, first paragraph: Please indicate the function of cyclosporin A in the main text, such as immunosuppressant medication, to help the readers better understand the experimental design.

The following changes were made:

Lines 126-127 “Cyclosporin A, a known immunosuppressant used to prevent transplant rejection” in the methodology section, page 3.

Lines 220-22 “Cyclosporin A (CsA), as a known immunosuppressant drug, was utilized to mitigate the effects of human PBMC transplant rejection and act as an active control for the delay of GvHD disease progression.”

Figure 2B: Why the weight of mice will increase firstly then decrease after the treatments of all PSCs and CsA? Do the authors have any explanation?

The mice utilized in the study were 6-8 weeks old. As part of the animal growth, the weight of the mice is expected to increase according to the mice strain (https://www.jax.org/jax-mice-and-services/strain-data-sheet-pages/body-weight-chart-005557).  The decline in weight is due to the onset of GvHD that is caused by the huPBMC administration and is expected as part of the disease progression.

Within methodology, an addition related to the mice age has been added in line 122 “6-8-weeks old”.

Figure 3A: What do the p-values mean in the figure? AM-PSC and PV-PSC can delay the time for the weight to return to Day 0 level more significantly?

There was an error in notation for statistical significance: only CsA and PV-PSCs demonstrated significantly slower weight loss over time. The onset of disease is marked by progressive loss in weight. As such, delay in weight loss is observed as % changed in weight compared to time 0 (baseline) of the study and represents a proxy measure of disease development. By using this metric, CsA and PV-PSCs treatments demonstrate the most uniform effect (statistical significance) in delaying the disease progression. The following information was added to reflect this point “However, only CsA and PV-PSC demonstrated the most uniform (statistically significant, p<0.05) effect in delaying the disease progression.”, lines 245-246.

The error in significance level was addressed where only CsA and PV-PSC groups are denoted with “*” for statistical significance in Figure 3A.

Minor issues:

Page 5, Results part: Please change “Figure 1B & C” to “Figure 1B” and change “Figure 1D” to “Figure 1C”. Please change “>10%” to “< 10%”.

Made the changes as suggested at lines 203, 207 and 208 on page 5 of the manuscript.

Similarly, made the following change at line 312 “<10% or negative” instead of “negative (<5%)” in the Discussion section, page 11.

Figure 3: I would recommend using the same order of different experimental groups for the legends in different panels, such as, “PBMC, CsA, PV-PSC, WJ-PSC, AM-PSC”.

 We have made changes to the legend of the figure as suggested for Figures 1-3.